# The Physiological Response of Different Brook Willow (*Salix acmophylla* Boiss.) Ecotypes to Salinity

**DOI:** 10.3390/plants11060739

**Published:** 2022-03-10

**Authors:** Emily Palm, Joshua D. Klein, Stefano Mancuso, Werther Guidi Nissim

**Affiliations:** 1Dipartimento di Scienze e Tecnologie Agrarie, Alimentari, Ambientali e Forestali (DAGRI), University of Florence, Viale Delle Idee 30, 50019 Sesto Fiorentino, Italy; emilyrose.palm@unifi.it (E.P.); stefano.mancuso@unifi.it (S.M.); 2Department of Natural Resources, Institute of Plant Sciences, Agricultural Research Organization, The Volcani Center, P.O. Box 15159, Rishon LeZion 7505101, Israel; vcjosh@agri.gov.il; 3Department of Biotechnology and Biosciences, University of Milano-Bicocca, Piazza Della Scienza 3, 20126 Milano, Italy

**Keywords:** willow, salinity stress, phytoremediation, *Salix acmophylla*, salt tolerance, drought tolerance

## Abstract

Few phytoremediation studies have been conducted under semi-arid conditions where plants are subjected to drought and/or salinity stress. Although the genus *Salix* is frequently used in phytoremediation, information regarding its tolerance of drought and salinity is limited. In the present study, *Salix acmophylla* Boiss. cuttings from three sites (Adom, Darom and Mea She’arim) were tested for tolerance to salinity stress by growing them hydroponically under either control or increasing NaCl concentrations corresponding to electrical conductivities of 3 and 6 dS m^−1^ in a 42-day greenhouse trial. Gas exchange parameters, chlorophyll fluorescence and concentration, and water-use efficiency were measured weekly and biomass was collected at the end of the trial. Root, leaf and stem productivity was significantly reduced in the Adom ecotype, suggesting that Darom and Mea She’arim are the more salt-tolerant of the three ecotypes. Net assimilation and stomatal conductance rates in salt-treated Adom were significantly reduced by the last week of the trial, coinciding with reduced intrinsic water use efficiency and chlorophyll a content and greater stomatal aperture. In contrast, early reductions in stomatal conductance and stomatal aperture in Darom and Mea She’arim stabilized, together with pigment concentrations, especially carotenoids. These results suggest that Darom and Mea She’arim are more tolerant to salt than Adom, and provide further phenotypic support to the recently published data demonstrating their genetic similarities and their usefulness in phytoremediation under saline conditions.

## 1. Introduction

The rapidly changing patterns in temperature and precipitation predicted by climate models over the next century are increasing pressure to change our approach to land use and land management [1,2]. It is expected that arid zones will expand [3] and levels of soil salinity will increase in certain areas [4], both of which will have significant negative impacts on agriculture and food production. Simultaneously, awareness of environmental pollution from various sources, including industrial production and consumer waste, has dramatically increased with the current changes in global climate patterns and the recent COVID-19 pandemic [5]. Changing climate regimes and increased soil pollution will likely reduce the amount of arable land available for agriculture unless steps are taken to reduce their impact [6,7].

There is growing interest in using green technologies to mitigate the negative impacts of pollution, including the plant-based approaches broadly termed phytotechnologies. Among the number of different strategies than can be adopted to enhance environmental protection, the implementation of phytotechnologies on a larger scale may represent a potent tool toward meeting this goal [8]. One of the more widely used and tested phytotechnologies is phytoremediation, that is, the use of plants and their associated soil microbiomes to ameliorate polluted soil. Phytoremediation has already been successfully used to address a substantially large number of environmental issues, including soil pollution [9,10] and the treatment of specific hazardous materials such as sewage sludge [11], landfill leachate [12,13], and municipal wastewater [14,15].

Despite its obvious potential, most phytoremediation techniques have been implemented in temperate regions, while other climates, especially those that are semi-arid, have received very little attention. In these environments plants are often subjected to multiple simultaneous constraints such as drought, soil salinization, erosion, and high temperatures that limit growth and other crucial physiological processes [16]. In response to these constraints, researchers have shifted the focus toward evaluating the performance of different phytoremediation approaches that utilize highly salt-tolerant halophyte species [17], as these often demonstrate a broad range of tolerance mechanisms that allow them to colonize different semi-arid ecosystems [18,19]. In this context, plants selected for phytoremediation in semi-arid regions would ideally show both tolerance to stressful environmental conditions and the capacity to ameliorate soil quality.

Willow is often cited as an ideal candidate genus for phytoremediation because it possesses many of the desirable traits for this purpose: fast rates of growth and harvestable biomass production [20,21], relatively high tolerance to many common organic and inorganic soil contaminants [22], and the availability of cultivars, hybrids and clones that are adapted to a wide range of climate and pedological conditions [23]. There are an estimated 330–500 species of willow (genus *Salix*). They are generally found in both temperate and arctic areas in the Northern Hemisphere [24]. These include some species that, because they grow in semi-arid regions, are potentially suitable for use in phytotechnology either in saline environments or for the treatment of saline materials (e.g., wastewater, landfill leachate). Information about willow drought stress tolerance is abundant, ranging from physiological [25,26,27] to more ecologically-focused studies [28,29]. However, relatively little research has been conducted on salinity tolerance in willow. A recently published study carried out in the North American prairie region comparing different commercial willow hybrids and native willow species has highlighted that native species are more tolerant than commercial hybrids to increased soil salinity [30]. The same conclusions were drawn by other scientists who compared two species, *Salix linearistipularis* (Franch.) K.S. Hao and *Salix matsudana* Koidz. in northeast China and observed that their actual natural habitat distribution reflected their differing degrees of soil salinity tolerance [31].

One willow species, *Salix acmophylla* (Boiss), is a particularly strong candidate for phytotechnologies in semi-arid climates given its wide-ranging natural distribution and its demonstrated tolerance to heavy metals [32]. *S. acmophylla* belongs to the section *humboldtianae* and is a medium-sized tree or shrub distributed in arid tropical and subtropical continental regions in the Middle East and Central Asia, where it grows along riverbanks and at altitudes including mountainous regions. Traditionally, *S. acmophylla* has been used in Central Asia as an energy source and for building materials in the Himalayan region as well as for the production of textiles such as basketry and the dying of carpets [33]. More recently, this species has been successfully used as green fodder for goats when irrigated with pre-treated municipal wastewater containing high concentrations of salts [34]. This green fodder provided specific chemicals that may be of significant value in maintaining and/or improving the health and welfare of animals [35]. The suitability of *S. acmophylla* for use in phytoremediation under saline conditions has not yet been explored, nor has its tolerance to salinity stress in general. In the present study, the physiological effects of increasing salinity stress on ecotypes of *S. acmophylla* collected from three different sites in Israel were evaluated under controlled greenhouse settings. We hypothesized that cuttings collected from areas with lower annual precipitation would be more physiologically adapted to salinity stress as demonstrated by greater biomass production and greater water use efficiency relative to cuttings sourced from comparatively wetter environments.

## 2. Results

### 2.1. Gas Exchange Parameters

During the first week under the electrical conductivity (EC) treatment of 3.0 dS m^−1^ (30 mM NaCl), no significant differences in net assimilation rate (A*_N_*) were observed between the three *Salix* ecotypes. However, from the second week (T2) and up to the end of the trial (T4), the ecotypes differed in response to salinity (Figure 1a), with significant main effects of ecotype (*p* < 0.022) and treatment (<0.001) and significant interactions between the two factors (*p* = 0.004) (Table 1). By the end of the second week of exposure to EC = 3 dS m^−1^, Mea She’arim and Darom had no significant change in A*_N_*, while the A*_N_* of Adom significantly decreased in response to salinity compared with control (14.1 vs. 21 µmol CO_2_ m^−2^ s^−1^, respectively). When subjected to higher EC, Darom was the only accession of which the A*_N_* was not significantly affected by salinity. In contrast, both Adom and Mea She’arim showed a reduction in A*_N_* in response to salinity relative to control-grown plants, though the reduction was more drastic in Adom (i.e., 10.9 vs. 16.3 µmol CO_2_ m^−2^ s^−1^ and 6.5 vs. 13.6 µmol CO_2_ m^−2^ s^−1^, respectively, in T3 and T4) than in Mea She’arim (i.e., 15.5 vs. 18.6 µmol CO_2_ m^−2^ s^−1^ and 13.6 vs. 15.8 µmol CO_2_ m^−2^ s^−1^, respectively, in T3 and T4).

The g_s_ of Adom was the most sensitive to the salinity treatment (Figure 1b). Before increasing the salinity of the solution, Adom had significantly higher g*_s_* rates than Mea She’arim and Darom, and again one week after adjusting EC to 3 dS m^−1^. By contrast, both Mea She’arim and Darom had a significant reduction in g_s_ in response to salinity that was observed up to the third week of the salt treatment (T3). By the end of the trial, only Adom exhibited a dramatic reduction in stomatal conductance under salinity stress compared to control, with a dramatic reduction overall between T0 and T4. The level of salinity was the only factor to have a strong effect on stomatal conductance rates (*p* = 0.002; Table 1)

Values of intercellular CO_2_ concentrations (C_i_) were highly variable among the ecotypes and between control and salt treated plants (Figure 1c), with a significant interaction between ecotype and treatment (*p* = 0.001; Table 1). Relative to control plants, C_i_ values of salt-treated Darom and Mea She’arim were equivalent or increased over time. In contrast, increasing the EC to 6 dS m^−1^ caused a significant reduction in C_i_ in Adom relative to control plants and relative to EC 3 dS m^−1^ treated plants. The intrinsic water use efficiency (WUE*_i_*) was mostly affected by the interaction between ecotype and salinity (Figure 1d), *p* = 0.001 (Table 1). This parameter did not change significantly among the three ecotypes throughout the trial under control (average 45 µmol mol^−1^). However, under salt stress Adom consistently had lower values of WUE_i_ than the other two ecotypes. The photosystems of Darom and Mea She’arim were generally more sensitive to salinity stress than those of Adom (Figure 2), although this did not seem to negatively impact growth (Table 1).

The maximum capacity of the photosystems (Fv/Fm) did not vary significantly among the ecotypes or in response to the increasing EC of the solution (Figure 2a; Table 1). However, by T4, an EC of 6 dS m^−1^ stimulated reductions in both PSII yield (Figure 2b) and efficiency (Figure 2c) in both salt-treated Darom and Mea She’arim, with a significant main effect of ecotype (*p* = 0.006 and *p* = 0.003, respectively; Table 1). The trends for electron transport rate (ETR; data not shown) mirror those of PSII efficiency, as the light intensity in the measurement chamber was the same for all samples. No such reductions were observed in Adom. Non-photochemical quenching (NPQ) did increase over time in both control and salt-treated plants from all three ecotypes; however, the highest values were found in salt-treated Mea She’arim at T4 (Figure 2d). There was a significant interaction between ecotype and treatment on values of NPQ (*p* = 0.044; Table 1).

### 2.2. Pigment Content

There were no significant differences in a, b or total chlorophyll concentrations under the low EC treatment (i.e., 3 dS m^−1^) among all three *S. acmophylla* ecotypes relative to the control (Figure 3). However, by the end of the first week at EC 6 dS m^−1^ (T3) the ecotypes demonstrated different responses to salinity, with Mea She’arim and Darom having a lower chl a concentration under salt stress compared with control and Adom (Figure 3a). By contrast, salinity did not significantly reduce either chl b (Figure 3b) or total chl (Figure 3c) concentrations in Mea She’arim or Darom, while it did negatively affect the same parameters in Adom. One additional week of treatment exacerbated these differences in response to salinity. As with the low EC treatment, there was no significant difference in chlorophyll (a, b and total) concentration in response to salinity in Mea She’arim and Darom, whereas Adom had a strong decrease in response to salinity. It is noteworthy that Darom had more total chlorophyll (2.20 mg g^−1^) compared to Mea She’arim (1.90 mg g^−1^) and Adom (1.92 mg g^−1^) throughout the trial. The significant increase in the chlorophyll a/b ratio observed in Adom starting at week three of the salt treatment and continuing to the end of the trial did not occur in either Darom or Mea She’arim (Figure 3d). The plant response to increasing salinity in terms of carotenoids depended very much on the accession (Figure 3e), with a significant main effect (*p* < 0.001; Table 1). Starting at T2, Mea She’arim and Darom had a general increase in total carotenoid concentration, while only slight and non-significant differences were observed in Adom. Mea She’arim had increases of 48.1%, 25.6%, and 24.4% in total carotenoids in response to salinity at the T2, T3 and T4 sampling points, respectively. Significant increases were observed in Darom as well, with salinity resulting in increased carotenoid concentrations of 76.1% (T2), 58.2% (T3) and 39.8% (T4) relative to control.

### 2.3. Stomatal Analyses

#### 2.3.1. Stomata Aperture

The degree of stomatal aperture at midleaf was primarily driven by the ecotype, rather than by treatment (Table 2; Appendix A). In fact, before salt treatments even started, the stomatal aperture of Adom was wider (58.9 μm^2^) than that of Darom (46.2 μm^2^) or Mea She’arim (36.1 μM^2^). Such differences remained throughout the duration of the trial. However, after salt stress was applied, interactions between ecotype and salinity immediately became significant at midleaf, and after two weeks at the tip as well (Table 2). In particular, we observed that at T1 salinity induced a significant 27.7% decrease in stomatal aperture in Darom but not in either Adom nor Mea She’arim. At T2, only Mea She’arim had a decrease in stomatal aperture following salinity (61.2 μM^2^ vs. 52.5 μM^2^). Starting at T3, Darom and Mea She’arim exhibited a decrease in stomatal aperture, while it remained unchanged in Adom. By the end of the trial (T4), all ecotypes had greater stomatal aperture under control than under salinity stress. Similar responses were observed at leaf tip level, with stomatal aperture in Adom consistently being greater than in other two ecotypes. In contrast to Adom, Darom and Mea She’arim had relative reductions in stomatal aperture due to salinity starting at T3. Surprisingly, at T4, a 32% increase in stomatal aperture under salinity was observed for Adom.

#### 2.3.2. Stomata Density

Unlike stomatal aperture, the density of stomata on both mid and tip abaxial leaf surfaces was not affected by either experimental factor (i.e., neither ecotype nor salt stress) (Table 3; Appendix A). We did observe a general increase in stomatal density over time for plants from all three ecotypes regardless the treatment; however, a clear response to salt stress was not found. The amount of open pore space was considered by multiplying the average aperture (µm^2^) by the stomatal density of the observed leaf area (cm^2^), as in Ginsberg and Klein, 2020 [36]. In all three ecotypes, the amount of open pore increased over time from T0 to T4 in both control and salt treated plants (data not shown); however, these differences were not statistically significant based on a repeated measures two-way analysis of variance.

### 2.4. Biomass Response

At the end of the trial, fresh and dry weight of the separated plant tissues (leaves, stems and roots) depended significantly (*p* < 0.05) on the interaction between ecotype and salinity treatment (Table 4 and Appendix A). The fresh weight of leaves, stems, roots and total biomass was significantly reduced by the salinity treatment only for Adom. While average stem biomass of Mea She’arim (32.2 g plant^−1^) and Adom (32.9 g plant^−1^) was greater than that of than Darom (22.3 g plant^−1^), the fresh root biomass of Darom was greater (45.7 g plant^−1^) than that of Mea She’arim (42.8 g plant^−1^) or Adom (37.3 g plant^−1^). Salinity significantly reduced leaf (26%) and root (24%) dry biomass compared to the control only in Adom. While the total plant dry biomass in Mea She’arim and Darom was not significantly affected by the treatment, it was significantly reduced (22%) in Adom under salinity stress compared with the control. Only in Darom was the root-to-shoot ratio significantly reduced in response to salinity, by 0.45 and 0.37 in control and salt-treated plants, respectively.

## 3. Discussion

Few of the published studies related to phytoremediation have been conducted in semi-arid conditions where high temperatures, limited water availability and increased soil salinity may be confounding factors [37]. It is relatively uncertain whether any of the genera of plants commonly used in temperate-zone phytoremediation, such as willow and poplar, might be suitable for similar projects in semi-arid regions. However, a common factor in phytoremediation is the increased electrical conductivity of landfill leachates and waste waters applied to soil or used hydroponically, which is due to a high level of dissolved salts [12,38,39]. This can affect the productivity of plants and alter their ability to absorb heavy metals such as cadmium and zinc [40,41]. Tolerance to salinity may therefore be an ‘indicator mechanism’ that can help determine whether further consideration of a potential phytoremediator is worthwhile.

In the present study, ecotypes of *S. acmophylla* from semi-arid regions in Israel were selected as candidates for salt-stress screening. *S. acmophylla* is adaptable to a variety of environmental conditions, including irrigation with wastewater [34]. The three ecotypes used in the present study were collected from three different sites that varied in geographic location and annual precipitation levels. Salt-treated plants were exposed to a low-intensity treatment of 3 dS m^−1^ (30 mM NaCl) for two weeks followed by two weeks with 6 dS m^−1^ (60 mM NaCl), a level of salinity similar to that found in most phytoremediation sites and wastewater. Root and shoot growth clearly demonstrated that ecotypes Mea She’arim and Darom were more tolerant to saline conditions than the third ecotype, Adom. The biomass data strongly indicate that adaptation to local environmental conditions is not the dominant factor determining tolerance to salinity stress. If this were the case, both Adom and Mea She’arim would likely be the more salt-sensitive ecotypes, and Darom, from the site with the lowest annual precipitation, would be the most salt-tolerant.

All our data effectively group Mea She’arim and Darom together and separate them from Adom based on their phenotypic responses to salt stress. At the time the cuttings were collected and the trial performed, it was thought that the populations represented three ecotypes adapted to specific site conditions. Genomic and chemical analyses subsequently demonstrated that Mea She’arim and Darom group together as hybrids of *S. acmophylla*, and potentially *S. alba*, given its ubiquitous occurrence throughout the region. The accession Adom is instead a part of a separate group of pure *S. acmophylla* clones found in both Israel and Jordan [42]. The three ecotypes represent two separate chemotype groups of secondary compounds, several of which may be important for abiotic stress responses. The phenotypic response observed here with Darom and Mea She’arim further supports the hypothesis that *S. alba* is a parent of these two ecotypes, which may explain their observed phenotypic tolerance to the higher salinity solution. *S. alba* was recently found to be salt tolerant up to 342 mM NaCl (or 34.2 dS m^−1^) through an enhanced ability to regulate Na translocation from the roots to the shoots [43].

The factors driving variations in productivity, and therefore sensitivity to salt conditions, among these three ecotypes may be found among the several photosynthetic and morphological parameters investigated in this study. Darom and Mea She’arim are potentially salt tolerant, which may be due to a combination of physical and biochemical adjustments related to the regulation of carbon assimilation rates. Net assimilation can be reduced by limitations in CO_2_ diffusion into the leaf (reductions in stomatal conductance) or through changes in leaf biochemistry. Stomatal conductance rates between the three ecotypes were not dramatically different in response to salinity stress. All three were mildly affected by the low-intensity salt treatment at the start of the experiment. While Mea She’arim and Darom stabilized despite an increase in the salinity from 3 to 6 dS m^−1^, stomatal conductance rates in Adom declined continuously; however, the difference between control and salt-treated plants was not as dramatic as seen with carbon assimilation rates. This conclusion is supported by the fact that there was no difference in stomatal aperture between control and salt-treated plants in the Adom ecotype, while in Mea She’arim and Darom, there was a decline in stomatal aperture with increased salinity. Furthermore, intrinsic water use efficiency was significantly higher in Mea She’arim and Darom in response to salinity relative to Adom, indicating a distinct difference in the ability to regulate stomatal aperture and a possible mechanism of salinity tolerance.

Regulation of stomatal aperture and density is a key factor in drought and salt tolerant species and ecotypes of major crops as a mechanism to reduce water loss from plant tissues [44,45,46,47]. Changes in water potential caused by water deficit or by increased soil salinity trigger ABA-driven stomatal closure. Studies with *Arabidopsis thaliana* [48] have shown that there is ecotypic variation in the efficiency of this signaling pathway, leading to differing degrees of tolerance to drought and salinity. While stomatal density is genetically determined, it can be influenced by environmental factors [49]. And though differences were not found among the plants in this trial, both moderate drought and salt stress can increase stomatal density [50]; on the other hand, with increasing duration and severity, decreases may be observed [51]. The level of salinity used or the duration of the present trial may not have been sufficient to stimulate changes in leaves emerging after the application of the stress. Furthermore, greater capacity to regulate stomatal aperture at the initial exposure to salt may be coupled to other mechanisms that assist in tolerance of salinity stress. Once Na enters the xylem of the roots, its rate of translocation to the shoots is regulated predominately by the transpiration stream [52]. Both Mea She’arim and Darom had a significant decrease in stomatal conductance rates one week after the application of the low-intensity salt treatment, although this decline subsequently stabilized. Reducing stomatal conductance may be a mechanism for reducing the entrance of Na ions through bulk flow driven by transpiration rates.

Several crop species upregulate key ion transporters for K^+^ in the roots when under sustained exposure to salt. This response helps overcome K^+^/Na^+^ imbalance in tissues, as maintenance of high cytosolic K^+^/Na^+^ is considered a key salt tolerance trait [43,53]. The SOS1 [54,55] and HKT [56] gene families have been reported to be involved in the extrusion of Na to the external media and removal of Na from the transpiration stream in the xylem, respectively. By preventing the translocation of Na^+^ to the shoots, K^+^/Na^+^ homeostasis in the leaves can be maintained, as was recently demonstrated in *S. alba* [43]. Expression levels of these genes increase over time with exposure to salt stress, and variations in expression levels between ecotypes coincide with varying degrees of salt tolerance. Capacity to exclude and regulate Na movement should be further explored as a potential mechanism explaining the tolerance to salinity observed in the genotype/chemotype represented here by Darom and Mea She’arim [42].

A biochemical explanation as well may be behind the reduction in the carbon assimilation rates and final biomass of the salt treated Adom plants. Diffusion of CO_2_ into the leaves in this case was not necessarily limited by mechanical differences in stomatal aperture. Instead, reductions in chlorophyll concentrations may have led to decreased capacity of the photosystems to absorb the light energy needed to drive the production of ATP and NADPH for carbon fixation. Through increased concentrations of sodium in tissues, salt stress is well documented to lead to stress responses such as the production of reactive oxygen species and subsequent reductions in photosynthetic pigments, carbon assimilation rates and biomass production [57,58]. Carotenoids can buffer the negative effects of reactive oxygen species on chlorophyll a and b concentrations by preventing their degradation [59,60]. Adom demonstrated a sensitivity to increased salinity, with reductions in concentrations of both chl a and b; however, the reduction in b was, as seen by the increase in the chl a/b ratio at T3 and T4. Carotenoids were found in greater concentrations in plants from all three sites in response to salt treatment. However, the differences in concentrations in Mea She’arim and Darom salt-treated plants relative to control were significantly higher than those of Adom, suggesting that this may be another salt-tolerance mechanism in these two accessions that is lacking in the more salt-sensitive Adom ecotype.

### Implications for the Use of Salt Tolerant S. acmophylla Hybrids in Phytoremediation

Salinity is often a factor taken into consideration in phytoremediation projects, especially those involving the treatment of waste waters emanating from water treatment facilities and leachate from landfills. Soils with an EC of 4 dS m^−1^ or higher are classified as saline soils [61]. In many of the studies evaluating phytoremediation potential for the amelioration of landfill leachates or wastewater, the solution applied in the trial has an EC of 3 dS m^−1^ or higher [12,62,63]. *Salix matsudana* demonstrated effective removal of heavy metals and even increased biomass production when leachate solutions with an EC of up to 3.3 dS m^−1^ were applied [12,64]. Not all willow species are suitable for phytoremediation of landfill leachates, as some previous studies involving willow species have shown lower salinity tolerance compared to the *S. acmophylla* ecotypes tested here. For example, the growth of *Salix amygdalina* L. was negatively affected by leachate solutions with electrical conductivity (EC) values higher than 3.0 dS m^−1^ [43]. In another recent study screening three native North American willow species (*S. discolor*, *S. eriocephala* and *S. interior*), only the latter was able to grow when subjected to a salinity level of 3.0 dS m^−1^ [65].

While several common woody plants used for phytoremediation (such as some poplar and eucalyptus hybrids) have demonstrated a certain level of tolerance to high salinity [66,67], willow tolerance to saline environments has been less frequently investigated. The present study may support the findings of Muklada et al. [42] regarding the parentage of Darom and Mea She’arim as a result of hybridization between *S. acmophylla* and *S. alba*. The relatively high salt tolerance of Darom and Mea She’arim may therefore be the result of hybrid vigor. Both *S. acmophylla* and *S. alba* have demonstrated previous tolerance to increased EC [34,43]. Given the widespread use of willow species in phytoremediation, these two ecotypes show promise for use under saline conditions.

## 4. Materials and Methods

### 4.1. Plant Selection and Growing Conditions

We tested the effect of salinity stress in *Salix acmophylla* Boiss. with a hydroponic experiment. *S. acmophylla* was chosen from among the many species in this genus because of its plasticity to environmental conditions, including drought. Cuttings were collected in Israel from three sites, Darom 4 (referred to hereafter as Darom), Adom, and Mea She’arim, characterized by different annual precipitation and/or heights above sea level (a.s.l.). Darom is in the northern Negev desert (31°15′ N, 34°47′ E, 250 m a.s.l.), with an average of 300 mm rainfall per year, while Adom and Mea She’arim are from Motza (31°15′ N, 34°47′ E, 250 m a.s.l.) and Jerusalem (31°47′ N, 35°13′ E, 754 m a.s.l.), respectively, both with 520–540 mm rainfall per year (Appendix A). Plant material (30 cm long cuttings of stem tissue) was harvested in winter 2020 from potted plants originating from trees grown after collecting campaigns in 2003 (Adom and Mea She’arim) and 2010 (Darom), and stored at 2 °C following their collection before planting as unrooted woody cuttings.

The trial was carried out in a greenhouse with average day/night temperatures during the experimental period of 27/18 °C, natural light (with the intensity reaching 800 μMol m^−2^ s^−1^ on days of full sun), and an average humidity of 65 ± 5%. The hydroponic system consisted of a set of 5 L black polyethylene buckets, each provided with an aeration pump to supply air to the root system. The surface of each bucket was covered with a black plastic lid in which five willow cuttings were vertically anchored with foam plugs, with ¾ of their length submerged in the solution and ¼ above the lid. All cuttings were rooted for two weeks in ¼-strength Hoagland solution with electrical conductivity (EC) = 0.5 ± 0.04 dS m^−1^ [68]. By day 14, roots and shoots had emerged on all the cuttings. At this point, the cuttings were randomly and evenly assigned to one of two groups: control and salt treatment. Plants in the salt treatment group were subjected to the first salinity increase (EC = 3.0 ± 0.11 dS m^−1^; 30 mM NaCl) on day 14. On day 28, the salinity for the salt-treated group was further increased to EC = 6.0 ± 0.12 dS m^−1^ (60 mM NaCl). The prescribed salinity levels of the experimental growth solutions were obtained by adding NaCl to the ¼-strength Hoagland solution. Both control and salt treatment solutions were changed every 7 days throughout the trial. The experimental layout consisted of a completely randomized design with two treatment factors (ecotype, salinity) and five replicates each. The entire trial was conducted over a total of 42 days, consisting of 14 days to establish cuttings and 28 days of salt treatment.

### 4.2. Gas Exchange and Fluorescence Assessment

Gas exchange and fluorescence parameters were evaluated five times throughout the trial: T0 = day 14, before the start of the salt treatment (EC = 0.5 dS m^−1^); T1 = day 21, one week after EC = 3 dS m^−1^; T2 = day 28, two weeks after EC = 3 dS m^−1^, T3 = day 35, one week after EC = 6 dS m^−1^ and T4 = day 42, two weeks after EC = 6 dS m^-^^1^. Measurements were taken using a Licor 6400XT open system and a Leaf Chamber Fluorometer (Part No. 6400-40) (Licor Biosystems, Lincoln, NE, USA). One recently fully expanded leaf on each plant was selected prior to the measurement day and completely covered in aluminum foil to maintain the dark-adapted state at the start of measurements the following morning. All measurements were conducted between 09:00 and 12:00. Conditions within the measurement chamber were set at 400 µmol CO_2_, block temperature 25 °C, relative humidity 40–50%, and a light intensity of 1000 µmol m^−2^ s^−1^ with 10% blue for light-adapted measurements. The effect of salinity stress on the capacity and efficiency of photosystem II was evaluated through dark- and light-adapted measurements of fluorescence [69]: maximum PSII quantum yield (i.e., Fv/Fm; where F_v_/F_m_ = F_m_ − F_o_/F_m_, maximum (F_m_) and minimum (F_o_) fluorescence values under dark-adapted conditions), the efficiency of open reaction centers (F′_v_/F′_m_, where F′_v_/F′_m_ = F′_m_ − F′_o_/F′_m_, maximum (F′_m_) and minimum (F_o_) fluorescence values under light-adapted conditions), and effective quantum yield (ɸPSII, where ϕPSII = F′_m_ − F_s_/F′_m_, maximum (F′_m_) and steady-state (F_s_) fluorescence values under light-adapted conditions). The measurements of CO_2_ assimilation rate (A*_N_*), stomatal conductance rate, intrinsic water-use efficiency, and photosystem II efficiency in light-adapted leaves were performed following the application of 1000 µmol m^−2^ s^−1^ light and once steady-state values were reached for each of the parameters listed. Intrinsic water use efficiency (WUE_i_), defined as the amount of carbon assimilated per unit of water lost over a given area of leaf and unit of time, was calculated as the ratio of net carbon assimilation rate and stomatal conductance rate [70]. Values for non-photochemical quenching (NPQ) of photosystem II using dark- and light-adapted maximum fluorescence values (NPQ = F_m_ − F′_m_/F′_m_, where F_m_ and F′_m_ are as previously described) were obtained using the Licor 6400XT system.

### 4.3. Pigment Concentration

Chlorophyll a and b, total chlorophyll, a:b ratio and total carotenoid concentrations were evaluated from leaf tissue collected following each gas exchange measurement after the salt treatments were started (T1, T2, T3 and T4). Samples were not collected at T0 in order to limit the amount of destructive sampling performed early in the trial when plants had developed few leaves. Leaf tissue samples were collected from the same mid-leaf position on the same leaves used for gas exchange/fluorescence measurements using a 0.78 cm^2^ cork borer. The samples were immediately weighed to obtain FW for calculation of pigment content and frozen in liquid nitrogen. All samples were stored at −80 °C until the end of the trial, when they were processed simultaneously [71]. Frozen leaf discs were individually ground in 80% acetone on ice in the dark to a homogeneous suspension and centrifuged at 6000 rpm (Eppendorf Centrifuge Model no. 5415 D; Hamburg, Germany) until a pellet formed (~five min). Absorbance values were read at wavelengths 661, 646, and 470 for chlorophyll a, chlorophyll b and total carotenoid content, respectively, using a spectrophotometer (Biorad SmartSpec Plus, Hercules, CA, USA). Pigment concentrations were calculated as mg pigment per gram of leaf (FW) using the following equations described in Wellburn [72] for an 80% acetone extraction:C_a_ = 12.21A_663_ − 2.81A_646_(1)
C_b_ = 20.13A_646_ − 5.03A_663_(2)
C_x+c_ = (1000A_470_ − 3.27C_a_ − 104C_b_)/198(3)

### 4.4. Stomatal Analyses

Stomatal density and aperture were evaluated over the course of the trial by epidermal imprints following weekly gas exchange measurements. Once a gas exchange measurement was complete, a small drop of transparent acrylic lacquer was lightly spread onto abaxial leaf surfaces at the leaf tip and mid-leaf section where gas exchange measurements were performed. After the lacquer had dried to the touch, a piece of clear tape was used to remove the imprinted acrylic layer and attach it to a microscope slide. Photos of the leaf surface imprints were taken with a Zeiss SteREO Lumar V.12 fluorescence stereomicroscope (Carl Zeiss Microimaging GmbH, Jena, Germany) with Zeiss AxioCam camera, model MRm 1.4 MP CCD. Images were preprocessed using the AxioVision Imaging System software (Carl Zeiss Microimaging GmbH, Jena, Germany, version 4.9), adjusting the contrast to improve definition of guard cells and stomatal aperture. Images were captured at 200× magnification. Using the NIH open-source image analysis software Image J (Version 1.53a), stomatal density and aperture size were accessed at two locations (tip and mid-leaf) on abaxial surfaces at the five time points during the trial (T0, T1, T2, T3 and T4) when gas exchange measurements were performed.

### 4.5. Growth Measurements and Biomass

At the end of the 42-day trial, plants were removed from the hydroponic solutions and the roots, stems and leaves were separated, weighed immediately to obtain fresh weight, and dried at 70 °C. Once the samples had reached a constant value, the final dry weight was recorded. The root-to-shoot ratio, which represents one of the mechanisms by which plants cope with environmental constraints [73], was calculated by dividing the root dry weight by the combined aboveground dry biomass (stem + leaves) on five plants for each treatment (i.e., ecotype and salinity) combination.

### 4.6. Statistical Analysis

For all variables, the normality of distributions was checked with the Shapiro–Wilk test. Data that were not normally distributed were log-transformed prior to any further analysis. A repeated measures two-way ANOVA was performed on gas exchange, chlorophyll fluorescence and concentrations, and stomatal parameter data in order to test for statistically significant differences among ecotypes and treatments using the statistical analysis software SSPS (Version 24; IBM^®^ Armonk, NY, USA). Biomass data were analyzed by two-way ANOVA. In both cases, Tukey’s HSD was used for post hoc comparisons of means for each combination of ecotype and treatment within individual time points. Data presented are the means of five individual analyses with standard error.

## 5. Conclusions

To expand the application of phytoremediation techniques to semi-arid zones, plants that can withstand multiple environmental stresses such as high temperatures and salinity will need to be identified through rigorous screening trials. Willow (*Salix* spp.) as a genus is already recognized for wide-ranging phenotypic plasticity for both drought [74] and heavy metal toxicity [75]. Adaptation to local environmental conditions, such as water availability, was not necessarily predictive of tolerance to salinity in the present study as was originally hypothesized. This is in contrast to the results of a study by Feng et al. [31] in which *S. linearistipulans*, a naturally occurring species adapted to saline-alkali soils in northeast China, showed a higher level of tolerance to high concentrations of NaCl (200 mM) than *S. matsudana*, a species distributed over a wide range of habitats. In that instance, habitat conditions likely played a dominant role in determining tolerance to salinity between those two species. In the current study, the genetic component may have been the determining factor, with hybrids of *S. alba* being better adapted to saline conditions. Given that *S. alba* is found in a wide range of environmental conditions from temperate to semi-arid, pure lines and hybrids of *S. alba* may be suitable for the phytoremediation of soils in urban areas where toxic deicing compounds are applied in the winter [76]. As both *S. alba* and *S. acmophylla* have shown some degree of tolerance to saline treatments, a more encompassing screening trial including pure lines of both *S. alba* and *S. acmophylla* and hybrids with at least one or both of these parental genotypes would help to clarify the role that genetics plays in salt tolerance in willow.

## Figures and Tables

**Figure 1 plants-11-00739-f001:**
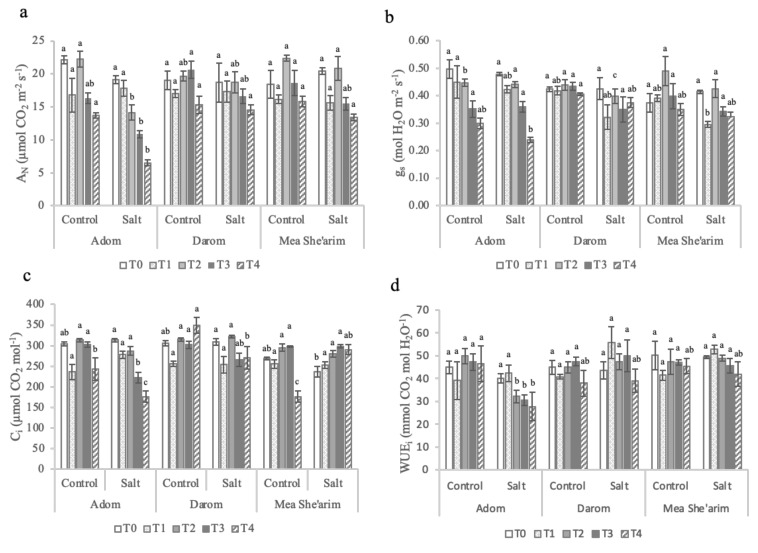
Weekly gas exchange measurements of the ecotypes Adom, Darom and Mea She’arim grown under control and saline conditions for 28 days. T0 = before the start of salt treatments; T1 and T2 = E.C. 3 dS m^−1^; T3 and T4 = E.C. 6 dS m^−1^. (**a**) A_N_ = net assimilation rate, (**b**) g_s_ = stomatal conductance rate, (**c**) C_i_ intercellular CO_2_ concentration, and (**d**) WUE_i_ = intrinsic water use efficiency. Bars represent the means of five replicates with SE bars. Different letters represent significant differences among ecotype and treatment combinations for individual timepoints following a post hoc Tukey HSD with a significance level of *p* < 0.05.

**Figure 2 plants-11-00739-f002:**
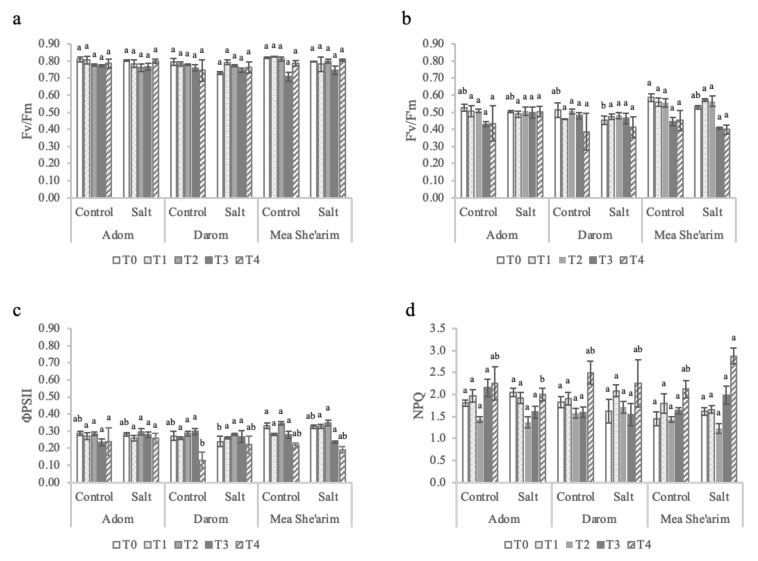
The effects of increasing salinity on photosystem II parameters. T0 = before the start of salt treatments; T1 and T2 = E.C. 3 dS m^−1^; T3 and T4 = E.C. 6 dS m^−1^. (**a**) Fv/Fm = dark-adapted maximum yield, (**b**) F′_v_/F′_m_ = light-adapted efficiency of open reaction centers of PSII, (**c**) ϕPSII = light-adapted effective quantum yield, and (**d**) NPQ = non-photochemical quenching. Bars represent the means of five replicates with SE bars. Different letters represent significant differences among ecotype and treatment combinations for individual timepoints following a post hoc Tukey HSD with a significance level of *p* < 0.05.

**Figure 3 plants-11-00739-f003:**
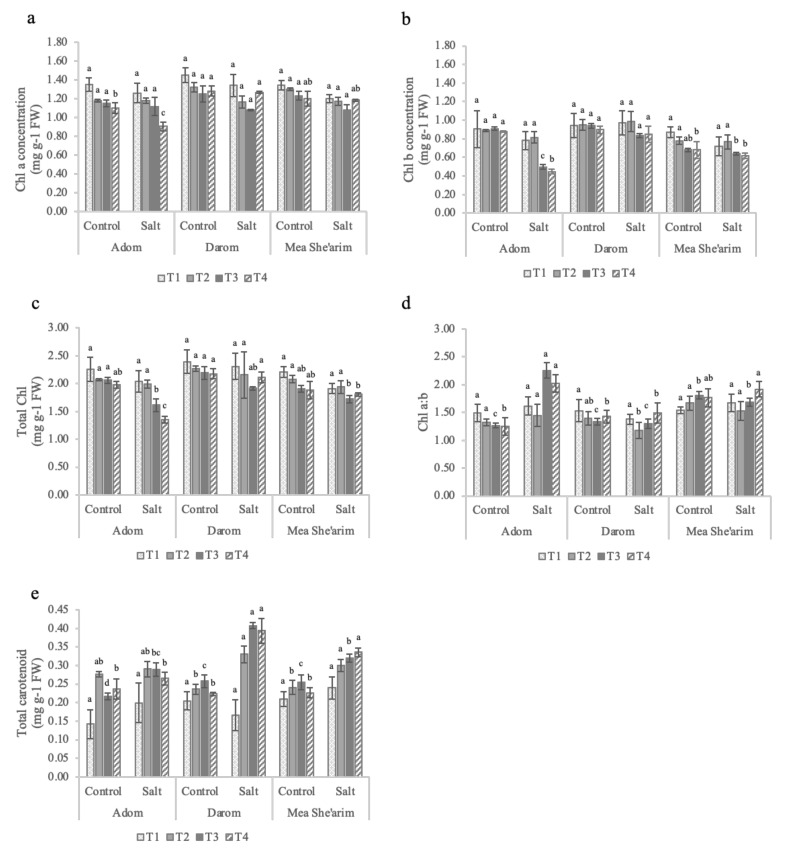
Weekly observations of leaf pigment concentrations in *S. acmophylla* ecotypes grown for 28 days in control and increasing salinity conditions. T0 = before the start of salt treatments; T1 and T2 = E.C. 3 dS m^−1^; T3 and T4 = E.C. 6 dS m^−1^. (**a**) Chl a = chlorophyll a, (**b**) Chl b = chlorophyll b, (**c**) total chlorophyll, (**d**) chlorophyll a/b, and (**e**) total carotenoids. All pigment concentrations were assessed as mg of pigment per gram fresh weight of leaf tissue. Bars represent the means of five replicates with SE bars. Different letters represent significant differences among ecotype and treatment combinations for individual timepoints following a post hoc Tukey HSD with a significance level of *p* < 0.05.

**Table 1 plants-11-00739-t001:** Repeated measures two-way ANOVA output for gas exchange (A_N_, g*s*, Ci and WUEi), fluorescence (Fv/Fm, F′v/F′m, ΦPSII and NPQ) and pigment (Chl a, Chl b, Total Chl, a/b and carotenoids).

Gas Exchange		A*_N_*	g_s_	Ci	WUEi	
	Ecotype (E)	0.002	NS	<0.001	0.002	
	Treatment (T)	<0.001	0.002	0.020	NS	
	E × T	0.004	NS	0.001	0.001	
Fluorescence		Fv/Fm	F′v/F′m	ΦPSII	NPQ	
	Ecotype (E)	NS	0.006	0.003	NS	
	Treatment (T)	NS	NS	NS	NS	
	E × T	NS	NS	NS	0.044	
Pigments		Chl a	Chl b	Total Chl	a/b	Carotenoids
	Ecotype (E)	<0.001	<0.001	NS	NS	<0.001
	Treatment (T)	<0.001	<0.001	NS	0.044	<0.001
	E × T	NS	NS	NS	NS	NS

Between-subject effects from repeated-measures two-way analysis of variance of main factors ecotype and treatment, and interactions. *p* values below 0.05 were considered significant.

**Table 2 plants-11-00739-t002:** Summary of stomatal aperture over the 28-day experiment growing *S. acmophylla* ecotypes under control and saline conditions.

Ecotype	Treatment	Stomata Aperture (µm^2^)														
		T0			T1			T2			T3			T4		
		Mean	SEM		Mean	SEM		Mean	SEM		Mean	SEM		Mean	SEM	
MID																
Adom	Control	56.8	*(3.5)*	b	53.5	*(6.3)*	b	61.8	*(3.6)*	b	66.3	*(6.1)*	a	61.5	*(4.5)*	a
	Salt	61.0	*(2.5)*	a	55.9	*(5.1)*	b	67.9	*(4.9)*	ab	66.9	*(4.6)*	a	55.9	*(2.5)*	c
	Mean	58.9	A		54.7	A		64.9	A		66.6	A		58.7	A	
Darom	Control	44.5	*(3.5)*	b	64.0	*(3.9)*	a	70.1	*(6.5)*	a	61.3	*(3.7)*	a	61.8	*(6.7)*	a
	Salt	47.8	*(3.7)*	b	46.5	*(3.2)*	bc	70.7	*(4.4)*	a	51.6	*(3.7)*	b	44.3	*(3.9)*	c
	Mean	46.2	B		55.3	A		70.4	A		56.4	B		53.1	AB	
Mea She’arim	Control	35.3	*(2.9)*	c	37.3	*(2.6)*	c	61.2	*(4.3)*	b	56.4	*(4.4)*	b	50.2	*(4.0)*	b
	Salt	36.9	*(4.2)*	bc	36.2	*(4.4)*	c	52.5	*(3.1)*	c	38.3	*(1.9)*	c	40.5	*(4.1)*	c
	Mean	36.1	C		36.8	B		56.9	B		47.4	C		45.3	B	
Ecotype (E)		<0.001			0.024			0.035			0.005			0.030		
Treatment (T)		NS			NS			NS			NS			0.029		
E × T		NS			0.045			0.023			0.043			0.040		
**TIP**																
Adom	Control	47.4	*(3.7)*		60.3	*(5.6)*		67.7	*(3.7)*		65.6	*(4.3)*	a	49.1	*(4.4)*	b
	Salt	51.8	*(2.6)*		65.7	*(3.7)*		67.2	*(5.5)*		65.0	*(5.3)*	a	65.1	*(4.1)*	a
	Mean	49.6	A		58.0	A		67.4			65.3	A		57.1	A	
Darom	Control	41.3	*(3.8)*		42.8	*(4.3)*		64.8	*(4.6)*		68.7	*(3.8)*	a	64.2	*(3.0)*	a
	Salt	44.8	*(3.2)*		49.8	*(3.3)*		54.8	*(5.0)*		46.5	*(2.9)*	c	54.3	*(5.0)*	b
	Mean	43.1	B		46.3	B		59.8			57.6	B		59.2	A	
Mea She’arim	Control	39.3	*(2.4)*		43.5	*(3.6)*		67.5	*(5.4)*		52.9	*(5.5)*	b	49.9	*(3.9)*	b
	Salt	41.2	*(3.2)*		46.5	*(4.1)*		55.2	*(4.0)*		35.0	*(1.2)*	d	42.4	*(4.3)*	c
	Mean	40.2	B		45.0	B		61.3			44.0	C		46.2	B	
Ecotype (E)			0.0134				0.0042				NS			<0.001		0.045
Treatment (T)			NS				NS				NS			0.001		NS
E × T			NS				NS				NS			0.045		0.023

T0 = before the start of salt treatments; T1 and T2 = E.C. 3 dS m^−1^; T3 and T4 = E.C. 6 dS m^−1^. Values for each factor are the means of five replicates ± SE. The results of a repeated measures 2-way NOVA demonstrate significant treatment and interaction effects with *p* < 0.05. Different letters reflect significant difference within individual timepoints, based on a post hoc Tukey HSD with a significance level of *p* < 0.05.

**Table 3 plants-11-00739-t003:** Summary of stomatal density over the 28-day experiment growing *S. acmophylla* under control and saline conditions.

Ecotype	Treatment	Stomata Density (n cm^−2^)									
		T0		T1		T2		T3	T4		
		Mean	SE	Mean	SE	Mean	SE	Mean	SE	Mean	SE
MID											
Adom	Control	173	*(23.0)*	152	*(13.6)*	222	*(22.2)*	226	*(20.5)*	198	*(9.7)*
	Salt	218	*(14.4)*	186	*(12.8)*	241	*(12.2)*	244	*(0.5)*	243	*(15.7)*
	Mean	196		169		231		235		221	
Darom	Control	161	*(23.2)*	190	*(30.1)*	205	*(12.3)*	199	*(14.7)*	192	*(19.0)*
	Salt	137	*(33.0)*	218	*(11.0)*	198	*(10.9)*	195	*(27.4)*	183	*(3.8)*
	Mean	149		204		201		197		188	
Mea She’arim	Control	151	*(9.8)*	173	*(7.2)*	203	*(12.2)*	211	*(25.0)*	193	*(11.8)*
	Salt	177	*(11.2)*	177	*(15.5)*	213	*(16.1)*	201	*(8.9)*	177	*(23.3)*
	Mean	164		175		208		206		185	
Ecotype (E)		NS		NS		NS		NS	NS		
Treatment (T)		NS		NS		NS		NS	NS		
E × T		NS		NS		NS		NS	NS		
**TIP**											
Adom	Control	192	*(13.5)*	175	*(11.4)*	203	*(13.5)*	213	*(12.2)*	209	*(11.3)*
	Salt	232	*(9.4)*	173	*(21.8)*	220	*(24.6)*	248	*(10.0)*	217	*(12.5)*
	Mean	212		174		211		231		213	
Darom	Control	158	*(28.6)*	227	*(12.5)*	222	*(13.4)*	161	*(13.1)*	238	*(14.4)*
	Salt	187	*(13.3)*	174	*(6.8)*	235	*(20.8)*	214	*(22.1)*	198	*(7.9)*
	Mean	173		201		228		187		218	
Mea She’arim	Control	173	*(10.9)*	180	*(33.0)*	209	*(11)*	207	*(16)*	222	*(18)*
	Salt	165	*(18.5)*	171	*(16.8)*	220	*(17)*	180	*(25)*	184	*(16)*
	Mean	169		175		214		193		203	
Ecotype (E)		NS		NS		NS		NS	NS		
Treatment (T)		NS		NS		NS		NS	NS		
E × T		NS		NS		NS		NS	NS		

T0 = before the start of salt treatments; T1 and T2 = E.C. 3 dS m^−1^; T3 and T4 = E.C. 6 dS m^−1^. Values for each factor are the means of five replicates ± SE. The results of a 2-way ANOVA demonstrate significant treatment and interaction effects with *p* < 0.05.

**Table 4 plants-11-00739-t004:** Biometric data (fresh weight (FW), dry weight (DW) and root-to-shoot ratio (R:S)) following 28 days of growth in control and increasing salinity conditions.

Ecotype	Treatment	Plant Biomass (g)																										
		Leaf						Stem						Root						Total						R:S		
		FW			DW			FW			DW			FW			DW			FW			DW					
Adom	Control	38.6	*(4.1)*	a	9.1	*(0.9)*	a	36.9	*(3.9)*	a	10.7	*(0.7)*	a	40.8	*(3.9)*	b	6.2	*(0.4)*	b	119.3	*(3.9)*	a	26.0	*(1.9)*	a	0.32	*(0.01)*	c
	Salt	29.2	*(2.0)*	b	6.7	*(0.5)*	b	28.8	*(2.3)*	ab	8.7	*(0.9)*	b	33.8	*(2.9)*	c	4.8	*(0.3)*	c	91.7	*(6.2)*	c	20.2	*(1.6)*	b	0.31	*(0.02)*	c
	Mean	33.9			7.9			32.9	A		9.7			37.3	B		5.5	B		105.5			23.1			0.32	B	
Darom	Control	38.5	*(2.6)*	a	6.0	*(0.3)*	b	27.4	*(5.1)*	b	10.4	*(0.6)*	a	47.5	*(2.4)*	a	7.4	*(0.2)*	a	109.3	*(2.3)*	ab	23.8	*(0.5)*	ab	0.45	*(0.01)*	a
	Salt	40.5	*(3.2)*	a	7.9	*(0.7)*	ab	17.2	*(3.1)*	*c*	8.7	*(0.8)*	b	43.9	*(2.8)*	ab	6.0	*(0.4)*	b	114.1	*(3.6)*	a	24.1	*(0.7)*	ab	0.37	*(0.03)*	b
	Mean	39.5			7.0			22.3	B		9.5			45.7	A		6.7	A		111.7			24.0			0.41	A	
Mea She’arim	Control	30.3	*(1.4)*	b	6.8	*(0.2)*	b	34.5	*(3.2)*	a	8.9	*(0.9)*	a	42.3	*(2.1)*	ab	5.2	*(0.2)*	c	102.8	*(4.3)*	b	20.9	*(1.2)*	b	0.34	*(0.02)*	bc
	Salt	37.9	*(3.0)*	ab	7.5	*(0.3)*	b	30.0	*(1.8)*	ab	7.6	*(0.9)*	b	43.2	*(2.3)*	ab	5.8	*(0.3)*	b	107.1	*(3.9)*	ab	20.8	*(1.1)*	b	0.39	*(0.03)*	b
	Mean	34.1			7.1			32.2	A		8.3			42.8	B		5.5	B		105.0			20.9			0.36	B	
ANOVA *p*-values																												
Ecotype (E)		NS			NS			0.012			NS			0.022			0.046			NS			NS			0.002		
Treatment (T)		NS			NS			NS			0.019			NS			0.003			0.091			NS			NS		
E × T		0.018			0.002			0.014			NS			0.015			0.002			<0.001			0.047			0.016		

Values for each factor are the means of five replicates ± SE. The results of a two-way ANOVA demonstrate significant treatment and interaction effects with *p* < 0.05. Significant differences between groups based on a post hoc Tukey HSD are indicated by different letters.

## Data Availability

All available data are contained within the article.

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
