# Peer review of "The Physiological Response of Different Brook Willow (Salix acmophylla Boiss.) Ecotypes to Salinity"

_plants, 2022, doi:10.3390/plants11060739_

Round 1
Reviewer 1 Report
In this study, authors evaluated the salinity tolerance of three accessions of S. acmophylla, which were collected from three different sites of Israel. Using several physiological parameters, they found that Darom and Mea She’arim are more tolerant to salt than Adom. Overall, the article is written clearly with adequate information in the introduction and methods. However, few things should be addressed.
- Authors should check the Na levels in three accessions, atleast in the root and discuss their possible mechanism of tolerance. This will also ensure their salt treatments in different aaccessions.
- Root biomass is different among accessions but I am wondering about the root length? These differences are due to root length or decrease in secondary root etc?
- How often authors change the media in hydroponic?
- I recommend authors adding a map which indicates the location of these accessions for clarity.
Author Response
Response to Reviewer 1 Comments
Point 1: Authors should check the Na levels in three accessions, atleast in the root and discuss their possible mechanism of tolerance. This will also ensure their salt treatments in different aaccessions.
Response 1: We fully agree that measurements of Na in the plant tissues would provide a more complete picture of the potential for tolerance among these three accessions. However, measurements of Na in the plant tissues were not performed originally because these analyses would need to be conducted by an external lab service and there was not sufficient funding dedicated to this project to pay for the digestion of tissues and the readings of the samples, even for just the root samples.
Point 2: Root biomass is different among accessions but I am wondering about the root length? These differences are due to root length or decrease in secondary root etc?
Response 2: This is an excellent question posed by Reviewer 1 but is not likely a robust factor to measure due to the fact that we were working with cuttings that developed adventitious roots. No clear single primary root emerged as it would with a seedling. Primary roots from seedling have been the source of most information collected on the effect of Na stress on root growth. Given the origin of our roots, and the fact that often several ‘primary’ roots emerge at the start of the trial, we did not make specific observations of root length or root system architecture. We fully agree that that sodium stress often does manifest first as reductions in primary root growth and this may lead to a stimulation in secondary root growth, or reductions in secondary root growth as well. We only considered root biomass as a whole in this trial as a measure of the effect of Na.
Point 3: How often authors change the media in hydroponic?
Response 3: The hydroponic solutions were changed once a week, following measurements of gas exchange/fluorescence. A statement has been added to the description of the experimental setup in Section 4.1.
Point 4: I recommend authors adding a map which indicates the location of these accessions for clarity
Response 4: A new map with the locations of differen eccotypes has now been provided in included in the supplementary materials.

Reviewer 2 Report
Emily Palm et. al. (plants-1614106) studied three willow ecotypes (Adom, Darom and Mea She’arim) under salt stress. The authors examined several physiological parameters.
I have some comments and suggestions for the Authors.
Line 105: EC: please specify the abbreviations when they first appear
I did not find the discussion of Figure 1b. in the text.
It would be advisable to move the Figure 1 before Table 1. It was not easy to follow and find Figure 1 on another page. I recommend to re-organize the position of figures and tables in the text for better readability.
If the Authors had taken photographs of the plants during the experiment, it would be sensible to upload them in a Supplementary file or as a figure to see what the plants looked like during the experiment, to understand the experimental design etc..
The reviewer works with salt stress, and compared to diagrams, a plain photo sometimes provides more information.
Line 259: Table 4. cannot be interpreted. Please re-organize this table.
Line 272: Figure 1. Please specify the significant level, and add value for it, p<0.05 is not enough. What do “a, b, etc” mean? Throughout the article, please check this.
Please add description to the panel a,b,c and d in the figure legend.
Line 293: Figure 2. Same as before. Please correct it.
I have one question: Why did not the Authors measured the ETR (electron transport rate) to monitor the photosynthetic ability of the stressed plants? This value is more informative compared to the Fv/Fm. Did the Authors measure this value?
Line 320: Figure 3. Same as before, please correct this.
Line 471: Plant material. If I understand right:
Authors wrote:
“Plant material (30 cm long cuttings of stem tissue) was harvested in winter 2020 from potted plants originating from trees grown after collecting campaigns in 2003 (Adom and Mea She’arim) and in 2010 (Darom) and stored at 2°C following their collection before planting as unrooted woody cuttings.”
Were the cuttings used in the experiment collected from plants grown in a greenhouse? Am I right? If so, how could the main question of the article be solved? :
Authors wrote in Line 99:
“We hypothesized that cuttings collected from areas with lower annual precipitation would be more physiologically adapted to salinity stress as demonstrated by greater biomass production and greater water use efficiency relative to cuttings sourced from comparatively wetter environments.”
Line 482: It would be desirable, if the NaCl concentration was also reported by the Authors, not just the EC value.
In Line 366 the Authors wrote: “342 mM NaCl (or 34.2 dS m-1)” from reference: 47
I’ve found the following calculation table:
https://plantstress.com/measuring-soil-salinity/
Solution EC (dS/m)
10 mM NaCl 1.0
This value correlates with what the Authors wrote in the text.
So, let’s see:
From Line 484:
EC=3.0 which means 30mM NaCl
EC=6.0 which means 60mM NaCl
Please correct me, if am I wrong.
I think, this NaCl concentration is too low. How and why did the Authors choose this very low NaCl concentration?
The reviewer works with Arabidopsis thaliana, which is a “salt sensitive” plant. For stress treatment, we usually use higher concentration of NaCl both in vitro or in vivo, even under short or long stress treatments. Please look at the reference 47:
Ran, X., Wang, X., Gao, X., Liang, H., Liu, B. & Huang, X. Effects of salt stress on the photosynthetic physiology and mineral ion absorption and distribution in white willow, 2021, PLoS One 6(11), e0260086.
In the referenced article, the Authors used: “NaCl concentrations of 171mmol 342mmol 513mmol 684mmol…”
I am not convinced that such low NaCl concentrations (30mM or 60 mM NaCl) used by the Authors could model a desert (Negev desert) environment. What can the Authors say about that?
Author Response
Response to Reviewer 2 Comments
Point 1:. Row 2. The physiological response of different brook willow (Salix acmophylla Boiss.) accessions to salinity. I suggest changing the title to: The physiological response of different brook willow (Salix acmophylla Boiss.) to salinity or The physiological response of different brook willow (Salix acmophylla Boiss.) ecotypes to salinity
Response 1: Following the reviewer’s suggestion we decided to change the title to ‘The physiological response of different brook willow (Salix acmophylla Boiss.) ecotypes to salinity’
Point 2:. Row 22 and many others: How to understand "accession". Is it a phenotype? or the place of origin of the phenotype? or its source? Or better fits ecotype? I think that the authors should change the word accession as the term accession is associated with joining or taking part in something.
Response 2: We decided that in this particular instance, ecotype does fit the situation better than accession. The plants from which the cuttings were sampled come from different locations with different environmental conditions. Additionally, we now know that there may in fact be a genetic component that accounts for the increased tolerance to the salt levels applied here in the plants from Darom and Mea She’arim. All instances of accession have been changed to ecotype.
Point 3:. Rows 39-44. Authors may consider the suggested literature position. Plants resistant to salinity will also be sought in other regions of the world other than semi-arid. Selection of appropriate vegetation, including Salix sp. varieties resistant to salinity, but also to soil contamination, e.g. heavy metals is also important in relation to urbanized areas and the vicinity of communication routes, where in winter sodium, potassium, and calcium chloride de-icing agents are used. This applies, of course, to climatic areas with snowy winters. Rolka E., Żołnowski A.C., Sadowska M.M. 2020 Assessment of Heavy Metal Content in Soils Adjacent to the DK16-Route in Olsztyn (North-Eastern Poland) Pol. J. Environ. Stud. 29(6): 4303-4311. https://doi.org/10.15244/pjoes/118384
Response 3: We agree with Reviewer 2’s suggestion to incorporate the possibility of using Salix varieties that are salt tolerant in general for other applications. That was idea behind this trial – choosing an EC similar to those found in phytoremediation studies of sewage sludge and landfill leachates that are often high in heavy metals. We added a sentence in the conclusion quoting the suggested paper.
Point 4:. Row 76 Why [25-26-27], it should be [25-27]
Response 4: This has been fixed as suggested by the reviewer.
Point 5:. Row 103 and 461 This question is for Editors: Is it possible to format the text as below
Material and methods
Results
Discussion
and so on, as it is in other publications in the MDPI group?
The reader should first learn about the experimental design and research methodology. In the methodology, the experimental treatment objects and abbreviations used later in the work are often explained. Only then should the results be discussed.
Response 5: For the Journal
Point 6:. Row 110: If you want to use the abbreviation Mea Sh, you should first define Mea She'arim (Mea Sh) and use Mea Sh in the rest of the text, see table 2. If you want to use Mea She'arim you have to change all Mea Sh abbreviations to Mea She'arim.
Response 6: The abbreviation was used just for the table due to formatting, but we have adjusted it to use Mea She’arim throughout the paper for consistency.
Point 7:. Row 332 Due to the rearrangement of the sections, the numbering order is not correct. The numbering of references does not match consecutive reference citations - it should be 36. Other numbers have to be checked again.
Response 7: This has been fixed as suggested by the reviewer. We have also added in the two references that were mentioned in the text but were not assigned numbers in the references section.
Point 8: Row 388 Why [48-49-50-51]? Not [48-51]
Response 8: This has been fixed as suggested by the reviewer.
Point 9: Row 452 Why [12-68-69]? Which is correct [12-69]; [12, 68-69]?
Response 9: The references should be 12, 68-69. This has been fixed as suggested by the reviewer.
Point 10: Row 461 see comment row 103
Response 10: This has been fixed as suggested by the reviewer.
Point 11: Row 483 The numbering of references does not match consecutive reference citations
Response 11: This has been fixed as suggested by the reviewer.
Point 12:. Row 474 In the remainder of the text, individual Salix trials from locations: Darom, Adom, and Mea She'arim are identified as ecotype Darom, ecotype Adom and ecotype Mea She'arim, respectively. May I suggest changing "accession" to "ecotype" or something similar?
Response 12: We decided to change all instances of ‘accession’ to ecotype as suggested by the reviewer. Using ecotype instead of accession is more consistent with the fact that these populations demonstrated different responses to the salt treatment applied in the present and come from locations with different environmental conditions.
Point 13:. Row 481-491 This part of the methodology is not understandable.
By day 14, seedlings were placed in a 25% Hoagland solution EC = 0.5 dS m-1.
Roots and shoots appeared.
On the 14th day, the seedlings were divided into groups:
Group no 1 - EC = 3 dS m-1
Group no 2 - EC = 6 dS m-1
I understand that in group 2 the salinity was raised to 3 dS cm-1 on day 14 and then to 6 dS cm-1 on day 28? Correct? Or straight to 6 dS on day 14?
Or maybe it was like this:
group 1 - control with constant EC level = 0.5 dS from first to the last day of the experiment.
group 2 - treated with 0.5 dS till 14 days, then increased EC to 3 dS on 14 day, then increased to 6 dS on 28 day.
Response 13: The second scenario described by Reviewer 2 is what was applied in the trial. We have altered the description to clarify that the plants started as cuttings were divided into two groups – a control group and a salt treated group. The salt treated group was supplied with a 3 dS m-1 solution at day 14 and then a 6 dS m-1 solution at day 28.
Point 14: Row 536 Can the authors provide the equations they used in calculating pigment concentrations? Reference literature is not available to the public, and readers who do not have access to Science Direct may not access this Wellburn publication [30]. Then he receives the message: Your selections could not be downloaded as you are not subscribed to that content.
The methodology provided should ensure that the achieved and described results can be verified in other conditions, such as e.g. different latitudes.
Response 14: We agree completely with the reviewer and have included the equations from the Wellburn 1994 reference.
Point 15: Row 551 If the authors have taken images, it is a good idea to include them in the work. Pictures taken showing the density of the stomata would be a very interesting improvement and the work would gain in value
Response 15: Representative photos of the epidermal peels for the ecotypes at all time points at each position (mid and tip) have been compiled into three new supplemental Figures (Sx, Sx and Sx) for Adom, Darom and Mea She’arim, respectively.
Point 16: 1. Somewhere between Row 613-620 Please insert suggested citation: Rolka, E., Żołnowski, A.C. & Sadowska, M.M. Assessment of Heavy Metal Content in Soils Adjacent to the DK16-Route in Olsztyn (North-Eastern Poland) 2020. Pol. J. Environ. Stud. 29(6): 4303-4311.You can find it here: https://doi.org/10.15244/pjoes/118384
Response 16: See point 2. The requested paper has been added

Reviewer 3 Report
Dear Authors
The manuscript submitted to review entitled: "The physiological response of different brook willow (Salix acmophylla Boiss.) accessions to salinity" I read with great interest. The research undertaken by the authors is quite a good fit with the publications presented in the “Plants” Journal. I rate manuscript relatively high. The authors noticed that global warming caused by climate change will result in deepening drought, which in semi-arid regions will be associated with an increase in soil salinity and the need to search for plant varieties resistant to salt stress. In their research, they focused on several Salix sp. ecotypes, which were tested for resistance to soil salinity.
I found a few inaccuracies which require important corrections. I have included all my comments in the pdf file attached as the reviewer comments
The most important observations that the authors should take under consideration are:
- Row 2. The physiological response of different brook willow (Salix acmophylla Boiss.) accessions to salinity.
I suggest changing the title to: The physiological response of different brook willow (Salix acmophylla Boiss.) to salinity
or
The physiological response of different brook willow (Salix acmophylla Boiss.) ecotypes to salinity - Row 22 and many others: How to understand "accession". Is it a phenotype? or the place of origin of the phenotype? or its source? Or better fits ecotype?
I think that the authors should change the word accession as the term accession is associated with joining or taking part in something. - Rows 39-44. Authors may consider the suggested literature position. Plants resistant to salinity will also be sought in other regions of the world other than semi-arid. Selection of appropriate vegetation, including Salix sp. varieties resistant to salinity, but also to soil contamination, e.g. heavy metals is also important in relation to urbanized areas and the vicinity of communication routes, where in winter sodium, potassium, and calcium chloride de-icing agents are used. This applies, of course, to climatic areas with snowy winters.
Rolka E., Żołnowski A.C., Sadowska M.M. 2020 Assessment of Heavy Metal Content in Soils Adjacent to the DK16-Route in Olsztyn (North-Eastern Poland) Pol. J. Environ. Stud. 29(6): 4303-4311. https://doi.org/10.15244/pjoes/118384 - Row 76 Why [25-26-27], it should be [25-27]
- Row 103 and 461 This question is for Editors:
Is it possible to format the text as below
Material and methods
Results
Discussion
and so on,
as it is in other publications in the MDPI group?
The reader should first learn about the experimental design and research methodology. In the methodology, the experimental treatment objects and abbreviations used later in the work are often explained. Only then should the results be discussed. - Row 110: If you want to use the abbreviation Mea Sh, you should first define Mea She'arim (Mea Sh) and use Mea Sh in the rest of the text, see table 2. If you want to use Mea She'arim you have to change all Mea Sh abbreviations to Mea She'arim.
- Row 332 Due to the rearrangement of the sections, the numbering order is not correct. The numbering of references does not match consecutive reference citations - it should be 36. Other numbers have to be checked again.
- Row 388 Why [48-49-50-51]? Not [48-51]
- Row 452 Why [12-68-69]? Which is correct [12-69]; [12, 68-69]?
- Row 461 see comment row 103
- Row 483 The numbering of references does not match consecutive reference citations
- Row 474 In the remainder of the text, individual Salix trials from locations: Darom, Adom, and Mea She'arim are identified as ecotype Darom, ecotype Adom and ecotype Mea She'arim, respectively.
May I suggest changing "accession" to "ecotype" or something similar? - Row 481-491 This part of the methodology is not understandable.
By day 14, seedlings were placed in a 25% Hoagland solution EC = 0.5 dS m-1.
Roots and shoots appeared.
On the 14th day, the seedlings were divided into groups:
Group no 1 - EC = 3 dS m-1
Group no 2 - EC = 6 dS m-1
I understand that in group 2 the salinity was raised to 3 dS cm-1 on day 14 and then to 6 dS cm-1 on day 28? Correct? Or straight to 6 dS on day 14?
Or maybe it was like this:
group 1 - control with constant EC level = 0.5 dS from first to the last day of the experiment.
group 2 - treated with 0.5 dS till 14 days, then increased EC to 3 dS on 14 day, then increased to 6 dS on 28 day. - Row 536 Can the authors provide the equations they used in calculating pigment concentrations? Reference literature is not available to the public, and readers who do not have access to Science Direct may not access this Wellburn publication [30]. Then he receives the message: Your selections could not be downloaded as you are not subscribed to that content.
The methodology provided should ensure that the achieved and described results can be verified in other conditions, such as e.g. different latitudes. - Row 551 If the authors have taken images, it is a good idea to include them in the work. Pictures taken showing the density of the stomata would be a very interesting improvement and the work would gain in value
- Somewhere between Row 613-620 Please insert suggested citation: Rolka, E., Żołnowski, A.C. & Sadowska, M.M. Assessment of Heavy Metal Content in Soils Adjacent to the DK16-Route in Olsztyn (North-Eastern Poland) 2020. Pol. J. Environ. Stud. 29(6): 4303-4311.
You can find it here: https://doi.org/10.15244/pjoes/118384
Please note other minor corrections in the attached pdf file
Yours sincerely
Reviewer

Author Response
Response to Reviewer 3 Comments
Point 1: Line 105: EC: please specify the abbreviations when they first appear
Response 1: This has been fixed as suggested by the reviewer.
Point 2:. I did not find the discussion of Figure 1b. in the text.
Response 2: Data already discussed in the text, but this was not made explicit by the authors by referencing Figure 1b. We apologize for the oversight. This has been fixed as suggested by the reviewer.
Point 3: It would be advisable to move the Figure 1 before Table 1. It was not easy to follow and find Figure 1 on another page. I recommend to re-organize the position of figures and tables in the text for better readability.
Response 3: We followed the format of the submission template. It is not clear to us whether we can move the order of tables and figures in this stage. We will be glad to do so and follow this sugegstion when authorized by the editor.
Point 4: If the Authors had taken photographs of the plants during the experiment, it would be sensible to upload them in a Supplementary file or as a figure to see what the plants looked like during the experiment, to understand the experimental design etc.. The reviewer works with salt stress, and compared to diagrams, a plain photo sometimes provides more information.
Response 4: Photos of plants before harvesting have been added to the supplementary materials
Point 5: Line 259: Table 4. cannot be interpreted. Please re-organize this table.
Response 5: As we are not sure how to change the content of Table 4 to make it easier to understand and did not receive feedback on this table from the other two reviewers, we did not alter the format. We will gladly take Reviewer 2’s concerns into consideration if more specific suggestions could be provided.
Point 6: Line 272: Figure 1. Please specify the significant level, and add value for it, p<0.05 is not enough. What do “a, b, etc” mean? Throughout the article, please check this.
Response 6: We cannot address this comment. The Figures report all information needed. Please read carefully the figure legend. Different letters represent significant differences among ecotype and treatment combinations for individual timepoints or treatments (depending on the paramerer considered) following a posthoc Tukey HSD with a significance level of p < 0.05.
Point 7: Please add description to the panel a,b,c and d in the figure legend.
Response 7: This has been fixed as suggested by the reviewer
Point 8: Line 293: Figure 2. Same as before. Please correct it.
Response 8: This has been fixed as suggested by the reviewer
Point 9: I have one question: Why did not the Authors measured the ETR (electron transport rate) to monitor the photosynthetic ability of the stressed plants? This value is more informative compared to the Fv/Fm. Did the Authors measure this value?
Response 9: The ETR was not included in the figure but was measured because it is a value that is automatically calculated by the Licor system. The values are therefore always available and can be provided. However, it was not included in the article or the figure because the values themselves are the result of multiplying the PSII efficiency by the light intensity. Given that light intensity is constant in the chamber for all measurements, the trends seen are the same as they would be for PSII efficiency and therefore add no additional information. Dark adapted values do not vary in this case, but that itself is still valuable information. A decision was made to prioritize data that provided unique information, but ETR can be included if there is space as determined by journal allowances.
Point 10: Line 320: Figure 3. Same as before, please correct this.
Response 10: This has been fixed as suggested by the reviewer.
Point 11: Plant material. If I understand right: Authors wrote: "“Plant material (30 cm long cuttings of stem tissue) was harvested in winter 2020 from potted plants originating from trees grown after collecting campaigns in 2003 (Adom and Mea She’arim) and in 2010 (Darom) and stored at 2°C following their collection before planting as unrooted woody cuttings.” Were the cuttings used in the experiment collected from plants grown in a greenhouse? Am I right? If so, how could the main question of the article be solved? Authors wrote in Line 99:“We hypothesized that cuttings collected from areas with lower annual precipitation would be more physiologically adapted to salinity stress as demonstrated by greater biomass production and greater water use efficiency relative to cuttings sourced from comparatively wetter environments.”
Response 11: If the mechanism for salt tolerance is genetically based, growing the plants from cuttings in a similar environment would not eliminate this capacity. The phenotype of salt tolerance would not necessarily be apparent under non saline conditions, but only when plants are exposed to saline conditions (see Palm et al., 2012). The results from this trial, in combination with the data reported in Muklada 2021 [42] suggest that the supposed parentage (Salix acmophylla (Adom) vs. Salix acmophylla x Salix alba (Darom and Mea She’arim) of these ecotypes may be responsible for variation in response to increased salt.
Point 12:. Line 482: It would be desirable, if the NaCl concentration was also reported by the Authors, not just the EC value
Response 12: We agree with the reviewer’s suggestion and have added the concentrations of NaCl following the EC values.
Point 13: In Line 366 the Authors wrote: “342 mM NaCl (or 34.2 dS m-1)” from reference: 47 I’ve found the following calculation table:https://plantstress.com/measuring-soil-salinity/Solution EC (dS/m)10 mM NaCl. This value correlates with what the Authors wrote in the text.So, let’s see:From Line 484:EC=3.0 which means 30mM NaClEC=6.0 which means 60mM NaClPlease correct me, if am I wrong.I think, this NaCl concentration is too low. How and why did the Authors choose this very low NaCl concentration?The reviewer works with Arabidopsis thaliana, which is a “salt sensitive” plant. For stress treatment, we usually use higher concentration of NaCl both in vitro or in vivo, even under short or long stress treatments. Please look at the reference 47:Ran, X., Wang, X., Gao, X., Liang, H., Liu, B. & Huang, X. Effects of salt stress on the photosynthetic physiology and mineral ion absorption and distribution in white willow, 2021, PLoS One 6(11), e0260086.In the referenced article, the Authors used: “NaCl concentrations of 171mmol 342mmol 513mmol 684mmol…”I am not convinced that such low NaCl concentrations (30mM or 60 mM NaCl) used by the Authors could model a desert (Negev desert) environment. What can the Authors say about that?
Response 13: The authors agree that relative to studies that are expressly studying the range of salt tolerance among different lines, cultivars species, etc, the level of salt used in this trial is low. However, we were interested in testing salinity levels consistent with those in phytoremediation studies using other willow cultivars. This would allow one to understand if tolerance to this level of salt could be a selection tool for the use of these willow varieties for phytoremediation of landfill leachate. Additionally we observed significant differences among our control and salt-treated plants despite these being relatively low levels of NaCl, and significant differences between Adom and the other two ecotypes Darom and Mea She’arim, suggesting that in this species, 60 mM NaCl is sufficient to see salt stress induced declines in various physiological parameters

Round 2
Reviewer 1 Report
authors answered all the queries and modified the text. Howver, they could not check the Na levels as i asked but i understand their reason.